

# Sparc: a sparsity-based consensus algorithm for long erroneous sequencing reads

Chengxi Ye[1] and Zhanshan (Sam) Ma[2]

[1] Department of Computer Science, University of Maryland, College Park, MD, USA
[2] Computational Biology and Medical Ecology Lab, State Key Laboratory of Genetic Resources and Evolution, Kunming Institute of Zoology, Chinese Academy of Sciences, Kunming, Yunnan, China

## ABSTRACT

**Motivation.** The third generation sequencing (3GS) technology generates long sequences of thousands of bases. However, its current error rates are estimated in the range of 15–40%, significantly higher than those of the prevalent next generation sequencing (NGS) technologies (less than 1%). Fundamental bioinformatics tasks such as *de novo* genome assembly and variant calling require high-quality sequences that need to be extracted from these long but erroneous 3GS sequences.

**Results.** We describe a versatile and efficient linear complexity consensus algorithm Sparc to facilitate *de novo* genome assembly. Sparc builds a sparse k-mer graph using a collection of sequences from a targeted genomic region. The heaviest path which approximates the most likely genome sequence is searched through a sparsity-induced reweighted graph as the consensus sequence. Sparc supports using NGS and 3GS data together, which leads to significant improvements in both cost efficiency and computational efficiency. Experiments with Sparc show that our algorithm can efficiently provide high-quality consensus sequences using both PacBio and Oxford Nanopore sequencing technologies. With only $30\times$ PacBio data, Sparc can reach a consensus with error rate <0.5%. With the more challenging Oxford Nanopore data, Sparc can also achieve similar error rate when combined with NGS data. Compared with the existing approaches, Sparc calculates the consensus with higher accuracy, and uses approximately 80% less memory and time.

**Availability.** The source code is available for download at https://github.com/yechengxi/Sparc.

Corresponding authors
Chengxi Ye, cxy@umd.edu
Zhanshan (Sam) Ma,
ma@vandals.uidaho.edu

## INTRODUCTION

Three generations of DNA sequencing technologies have been developed in the last three decades, and we are at the crossroads of the second and third generation of the sequencing technologies. Compared with the previous generations, the third generation sequencing (3GS) can provide reads in the range of 5–120 kilo-bases in one fragment. However, at present, the reported error rates are ~15% with PacBio sequencing (*Koren et al., 2012*), and
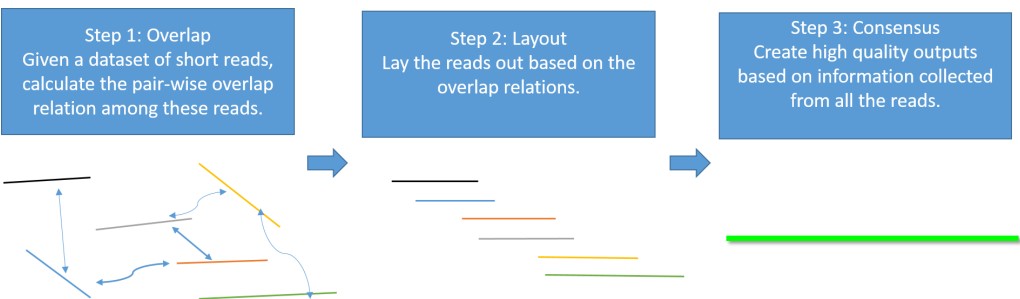

**Figure 1** A standard Overlap-Layout-Consensus genome assembly pipeline.

as high as ∼40% with Oxford Nanopore sequencing (*Laver et al., 2015*); this poses arguably the most significant computational challenge for assembling genomes with 3GS reads.

Genome assembly is the process of taking a large number of short and low-quality DNA sequences and putting them back together into contiguous pieces known as *contigs* to create a high-quality representation of the original chromosomes from which the DNA originated (*Myers et al., 2000*; *Nagarajan & Pop, 2013*; *Pevzner, Tang & Waterman, 2001*). With the 3GS data, *de novo* genome assembly algorithms need to pass through three major bottlenecks: finding overlaps (*Berlin et al., 2015*; *Ye et al., 2014*), sequence alignment (*Chaisson & Tesler, 2012*; *Myers, 2014*) and sequence polishing/error correction. Efficiently correcting these long erroneous reads is a non-trivial problem (*Au et al., 2012*; *Hackl et al., 2014*; *Koren et al., 2012*; *Salmela & Rivals, 2014*). Given a low-quality target genomic region, fast algorithms are required to collect the query sequences that can be aligned to the target region, and an accurate aligner is necessary to exploit the layout relations of the query sequences. Finally, a polishing/consensus algorithm takes the layout information to infer the "ground truth" sequence. Figure 1 summarizes the major challenges in a genome assembly pipeline, and the final step is the primary focus of this paper.

The consensus algorithm is critical for genome assembly for multiple reasons. Firstly, the consensus algorithm is a necessary part of an assembler to produce high-quality outputs. Secondly, recent assembly advancements resort to an error correction procedure (*Au et al., 2012*; *Hackl et al., 2014*; *Koren et al., 2012*; *Salmela & Rivals, 2014*) to raise the per-base accuracy in the input sequences. The consensus algorithm can also be used to polish each individual read; those corrected reads are usually used as high-quality inputs, and are fed into existing *Overlap-Layout-Consensus* based assemblers that require accurate inputs (*Huang et al., 2003*; *Mullikin & Ning, 2003*; *Myers et al., 2000*). In the first scenario, each draft assembly contig (or called backbone) is used as the target, and all the reads that align to this backbone are used as the query sequences to raise the quality of the backbone. In the second scenario, each long erroneous read is treated as the target; sequences from either NGS or 3GS may be used as the query sequences. Utilizing NGS data together with 3GS data, known as the hybrid assembly approach, has been a widely adopted assembly strategy since the birth of 3GS technologies. Since the NGS short read data cost significantly less and has higher accuracy, incorporating them can potentially reduce the cost and computational burden of the whole assembly pipeline (*Ye et al., 2014*). Nevertheless, due

to the lack of efficient consensus algorithm, this consensus step is usually circumvented by simpler approaches such as replacing regions in the target sequences with the NGS reads or assemblies. Unfortunately, errors in the NGS sequences may corrupt the originally correct 3GS sequences and create unwanted consensus errors in the final assembly. Finally, it is also noteworthy that the consensus step in genome assembly pipelines often takes the largest portion of the computational time (*Berlin et al., 2015*; *Chin et al., 2013*; *Lee et al., 2014*). Therefore, an efficient consensus algorithm can significantly accelerate the whole genome assembly process.

Traditionally, multiple sequence alignment, known to be a computationally challenging task is used to find the layout and construct a sequence alignment graph (*Edgar, 2004*; *Larkin et al., 2007*; *Lee, Grasso & Sharlow, 2002*; *Rausch et al., 2009*). Alignments are refined and clustered to infer the alignment profile as the consensus in the target region. The higher error rates lead to much higher complexities with these traditional approaches. To lower the complexity, researchers have tried to simplify the multiple sequence alignment by aligning all query sequences to a backbone sequence and creating a multigraph representing the alignment graph (*Chin et al., 2013*). Each nucleotide base leads to a graph node. Graph simplifications are applied to merge the multiple edges, and the best scored path is found as the consensus sequence. For NGS data, a similar strategy using the *de Bruijn* graph (DBG) has been developed (*Ronen et al., 2012*) to correct the assembly errors in single cell sequence assembly.

In this work, we borrow wisdom from the well-known *de Bruijn*/$k$-mer graph (*Hannenhalli et al., 1996*; *Pevzner, Tang & Waterman, 2001*; *Ronen et al., 2012*) and design a simpler graph formulation of the consensus problem for 3GS data. A general and versatile 'Sparc' algorithm is developed to polish long erroneous draft assembly sequences. Sparc builds a regular directed acyclic graph (i.e., non-multigraph) directly from the sequences. Each node in our regular graph is a $k$-mer. $k$-mers that appear in the same location are merged on the fly when constructing the graph to reduce memory consumption. The graph is also allowed to be 'sparse' (*Ye et al., 2012*) to further avoid using excessive memory caused by false $k$-mers. The links/edges between the $k$-mers are constructed when feeding in each read. Edge weight represents the reliability of the link. Intuitively, a path with the highest sum of edge weights is a good approximation of the consensus. This path is therefore searched and regarded as the algorithm output. Details can be found in the next section. Sparc can provide superb results at low memory, without utilizing any other graph simplification techniques. Due to its simplicity, the algorithm is five times faster and uses five times less memory space compared to a major 3GS consensus program PBdagcon (*Chin et al., 2013*). Moreover, since the accuracy of the prevalent NGS sequencing data (>99%) is significantly higher than that of the NGS data, it is desirable to use the cheaply available NGS to substitute some portion of the costly 3GS data. In this scenario, a consensus program capable of utilizing different types of data is highly anticipated and should provide higher quality output. Sparc is designed to meet these requirements and it provides high-quality results in the hybrid setting for major 3GS technologies.

*Sparc Consensus Algorithm*:

```
1.  Given the backbone sequence and k, g, build a
    position specific k-mer graph: sample every g
    k-mers, and record their location in the
    backbone sequence.
2.  Align each query sequence to the existing
    graph.
    2.1 If a query region suggests a novel
        path/variant then create a branch and
        allocate new k-mer nodes and links
        between these nodes.
    2.2 If a query region perfectly aligns to an
        existing region in the graph then
        increase the edges weights in the region.
3.  Reduce all the edge weights in the graph by
    max(c, t*cov).
4.  Use a Dijkstra-like breadth first search to
    search for a heaviest path as the final
    consensus sequence.
```

**Figure 2  The Pseudo-code of *Sparc*.**

## METHODS

Sparc consists of the following four simple steps: (i) Build an initial position specific $k$-mer graph (*Ye et al., 2012*) using the draft assembly/backbone sequence. (ii) Align sequences to the backbone to modify the existing graph. (iii) Adjust the edge weights with a sparse penalty. (iv) Search for a heaviest path and output the consensus sequence. The pseudo-code can be found in Fig. 2.

### Building the initial graph

Sparc uses $k$-mers to encode the local structure of the genomic region. It takes a preassembled draft assembly/backbone to build an initial $k$-mer graph (Fig. 3A). The $k$-mers in *different* positions of the backbone are treated as *independent* nodes. Therefore this initial graph is a linear list of the $k$-mer nodes. Note that this is a major difference from the popular *de Bruijn* graphs in genome assembly. Same $k$-mers in different locations are collapsed in a *de Bruijn* graph. To deal with the long reads, it is important to differentiate $k$-mers by their positions in the genome. In the consensus context, $k$-mers are position specific; different positions are treated *independently*. It is noteworthy that allocating $k$-mers in each location can take a large amount of memory, especially in the next stage. To circumvent this problem, we allow constructing a sparse $k$-mer graph (*Ye et al., 2012*) by storing a $k$-mer in every $g$ bases, which reduces the memory consumption up to $1/g$. We also record the edge links between the $k$-mer nodes. The edge weight represents the confidence in the corresponding path. We use coverage/multiplicity to represent confidence for simplicity. In the initial graph, the edge links have multiplicity 1, therefore edge weights are set to be 1. A generalization of using quality score to model the confidence of the links is straightforward.

### Aligning sequences to the backbone and building the whole graph

Sequences that align to the backbone sequence provide rich information about the ground truth sequence. Ideally, a most likely genome sequence should be searched as

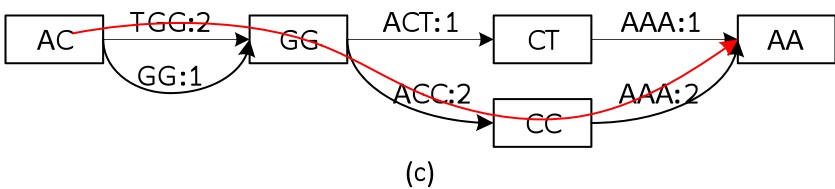

Backbone: ACTGGACTAAA, k=2, g=3

(a)

Backbone: ACTGGACTAAA
Seq1:        ACTGGACCAAA

Backbone: ACTGGACTAAA
Seq2:        AC_GGACCAAA

(b)

Find max path to build consensus sequence :
            ACTGGACCAAA

(c)

**Figure 3**   **A toy example of constructing the position specific sparse *k*-mer graph.**  (A) The initial *k*-mer graph of the backbone. (B) Adding two sequences to the graph. (C) The heaviest path representing the consensus is found by graph traversal (original weights are used in this example).

the consensus given all the input sequences. However, utilizing the multi-sequence information comprehensively requires computationally expensive operations such as pair-wise alignment of all the related sequences (*Edgar, 2004*; *Larkin et al., 2007*; *Lee, Grasso & Sharlow, 2002*; *Rausch et al., 2009*). Here we adopt a similar but simpler strategy in PBdagcon (*Chin et al., 2013*), by aligning all the sequences to the backbone, and modify the existing graph according to the alignments. Rather than creating an intermediate graph that needs to be refined or simplified (*Chin et al., 2013*; *Rausch et al., 2009*), we construct the final graph on the fly. We borrow the wisdom from constructing a *de Bruijn/k*-mer graph (*Pevzner, Tang & Waterman, 2001*; *Ye et al., 2012*): (i) If a query region suggests a novel path/variant, we create a branch and allocate new *k*-mer nodes and edges between these nodes. An example can be found in the upper half of Fig. 3B, when we align the last six bases of Seq1 to the existing graph. In this example, two new edges ACC and AAA with

multiplicity 1, and one $k$-mer node CC are allocated. (ii) If a query region perfectly aligns to an existing region in the graph, we increase the edges weights in the region without allocating new nodes. Examples can also be found in Fig. 3B. When aligning the first five bases of Seq1 to the existing graph, nodes AC, GG and edge TGG are merged implicitly with the ones created by the original backbone, the edge weights are increased by 1. When aligning the last six bases of Seq2 to the existing graph, the nodes and edges are merged implicitly with the ones created by Seq1, the edge weights are changed accordingly. As previously mentioned, this construction process shares similarity with the construction of a *de Bruijn* graph, but the nodes in our graph are differentiated by their $k$-mers and their positions in the backbone. In addition, Sparc is designed to facilitate hybrid assembly, and can leverage more weight to the high-quality data. When different types of sequencing data are available, higher weights can be assigned to the more reliable edges. The resulting $k$-mer graph contains rich information about the underlying genomic region. Next we describe another simple technique to extract the most likely sequence as the consensus output.

## Adjusting the weights of the graph

Intuitively, a path in the $k$-mer graph is likely to be genuine if it is supported by multiple sequences. Based on this intuition, a path with the highest confidence, i.e., the largest sum of edge weights should be a good approximate of the "ground truth" sequence. However, a direct search for this path may result in an erroneous output. A simple example is a long insertion error: the sum of weights of this erroneous path is higher even though there is only one supporting sequence. To circumvent this type of error, we subtract the edge weight by a small amount. This amount is adaptively determined with consideration of the sequencing coverage in each region. A fixed weight penalty is initially set to be $c = 1 \sim 3$; this shares the same heuristic with the parameter setting in existing *de Bruijn* graph assemblers. These assemblers remove false positive $k$-mers/links by detecting and removing infrequent $k$-mers/links in the dataset. We make this penalty adaptive to the sequencing coverage by introducing an adaptive threshold $t = 0.1 \sim 0.3$ of the average backbone multiplicity. The final penalty is the larger of the two. In all experiments, we fix these two parameters to be $c = 2$, $t = 0.2$. This penalty technique, also known as soft thresholding (*Mallat, 2008*), is equivalent to put an $l^1$-penalty on the edge weights. With this sparse penalty, the low coverage long insertion errors will be less likely to be favored compared to the genuine sequences.

## Output the heaviest path as the consensus

The graph constructed in section 'Aligning sequences to the backbone and building the whole graph' is a directed acyclic graph, and this allows us to search for a heaviest path with the adjusted weights. To do this, a *Dijkstra*-like breadth-first search is adopted to traverse the directed graph from the starting node to the ending node of the backbone. All edges before node $i$ in the backbone must be traversed before we start to traverse the edges starting from node $i$. For each node $i$, the heaviest path to it and its previous node are recorded. After the full traversal, a backtrack from the highest scored node in the backbone reports the highest scored path. This sub-path is output as the consensus path (Fig. 3C).

## Implementation details

The complexities of the above procedures described in all are linear to the data size. Currently Blasr (*Chaisson & Tesler, 2012*) is called to provide long read alignment information. Sparc takes a backbone file in fasta format and the Blasr alignment results as inputs. To avoid multiple placement of a query read, each read is mapped to one best matching region in the backbone. Reusing the consensus result as the input and iteratively running the consensus algorithm helps to improve the accuracy even more. Sparc allows for taking different $k$-mer sizes ($k$) and skip sizes ($g$). Constructing a sparse graph by skipping every $g$ $k$-mers is beneficial since it consumes $1/g$ memory and has a similar resolving power with a dense graph using $(k+g/2)$-mers (*Ye et al., 2012*). As in the *de Bruijn* graph, the $k$-mer size defines the anchor size when we analyze the alignments. A large number helps to distinguish between the repeats and avoids merging unrelated sequences. However, since the 3GS data is usually of high error rates, a smaller $k$-mer size ensures that the weaker alignments still share enough $k$-mers. A trade-off has to be made here and we find it is necessary to set this number smaller than in most *de Bruijn* graphs. In our experiments we have found that using $k = 1$–3, $g = 1$–5 is sufficient for practical purposes. When high-quality NGS data is available, the edges supported by the high-quality data shall be treated with higher priority. A parameter $b$ is introduced to selectively increase the weights of the reliable edges by a small amount ($b = 5 \sim 10$) to provide better cues for the correct path.

## RESULTS

Sparc has been tested on a variety of datasets. Here we demonstrate the test results from two PacBio datasets (http://schatzlab.cshl.edu/data/ectools/) and one Oxford Nanopore dataset (http://gigadb.org/dataset/100102). Sparc is designed to be a base-level consensus algorithm. While there are platform-specific ones that take into account signal processing-level information such as Quiver (for PacBio), and Nanopolish (for Oxford Nanopore), these programs usually take the outputs of the base-level ones as inputs to further improve the accuracy. As a fair comparison, we demonstrate results side-by-side with the most similar program to ours, which is PBdagcon (*Chin et al., 2013*). PBdagcon is the major module that is intensively used in HGAP (*Chin et al., 2013*) and MHAP (*Berlin et al., 2015*) pipelines to correct reads and generate consensus using base-level information. We therefore show the comparative results of both programs on these datasets. Both programs are fed with the same input data. We generate assembly backbones and collect the related reads for each backbone using DBG2OLC (*Ye et al., 2014*). Blasr (*Chaisson & Tesler, 2012*) is called (with option –m 5) to obtain the alignments. The final consensus error rates are calculated using the *dnadiff* function in MUMmer 3 (*Kurtz et al., 2004*). All experiments are conducted on a workstation with AMD Opteron 2425 HE CPUs (@ 800 MHz frequency). In some experiments we provide both NGS data and 3GS data. In these 'hybrid' settings 50× Illumina assembly contigs by SparseAssembler (*Ye et al., 2012*) are included and the edge weights are increased by $b = 5 \sim 10$. 50× Illumina data provides moderate coverage to allow off-the-shelf *de Bruijn* graph based assemblers to assemble high-quality contigs.

**Table 1  Results on an *E. coli* dataset using PacBio sequencing.**

| Program | Coverage | N50 | # | Time | Memory | Err1 | Err2 | Err4 |
|---|---|---|---|---|---|---|---|---|
| Sparc | 10× PB | 1.06 MB | 11 | 0.5 m | 308 MB | 1.95% | 1.51% | 1.50% |
| PBdagcon | 10× PB | 1.06 MB | 11 | 3.0 m | 1.10 GB | 1.95% | 1.52% | 1.51% |
| Sparc | 10× Hybrid | 1.06 MB | 11 | 0.5 m | 237 MB | 0.19% | 0.09% | 0.06% |
| PBdagcon | 10× Hybrid | 1.06 MB | 11 | 3.0 m | 1.23 GB | 1.02% | 0.64% | 0.58% |
| Sparc | 30× PB | 4.74 MB | 2 | 1.3 m | 2.30 GB | 0.41% | 0.16% | 0.11% |
| PBdagcon | 30× PB | 4.74 MB | 2 | 9.3 m | 7.70 GB | 0.49% | 0.23% | 0.18% |
| Sparc | 30× Hybrid | 4.74 MB | 2 | 1.3 m | 2.14 GB | 0.17% | 0.02% | 0.02% |
| PBdagcon | 30× Hybrid | 4.74 MB | 2 | 9.7 m | 9.58 GB | 0.49% | 0.18% | 0.13% |

**Table 2  Results on an *A. thaliana* dataset using PacBio sequencing.**

| Program | Coverage | N50 | # | Time | Memory | Err1 | Err2 | Err4 |
|---|---|---|---|---|---|---|---|---|
| Sparc | 20× Hybrid | 2.02 MB | 469 | 21 m | 1.7 GB | 0.36% | 0.19% | 0.17% |
| PBdagcon | 20× Hybrid | 2.02 MB | 469 | 123 m | 8.9 GB | 0.81% | 0.53% | 0.47% |

On PacBio datasets, we set $k = 1$, $g = 1$, and run the consensus algorithms for four rounds. The per-base error rates after 1/2/4 rounds are reported as Err1, Err2 and Err4 in Tables 1 and 2, respectively. In the first experiment, we use an *E. coli* PacBio dataset and test the accuracy using different sequencing coverages. The longest backbones generated by DBG2OLC using 10× / 30× data are 1.3 Mb and 4.6 Mbp respectively. The *E. coli* genome reference (4.6 Mbp) can be found with accession number NC_000913. An important assembly scenario is when we have both NGS and 3GS data, we find that Sparc facilitates this hybrid assembly approach and leads to both cost efficiency and computational efficiency. After two rounds, Sparc reaches an error rate of 0.09% using only 10× data in a hybrid setting compared to 0.64% using PBdagcon. As expected, the quality is even better with 30× data (0.02%). The error rates of using PacBio (PB) data only (i.e., non-hybrid) are higher as expected. Running the algorithms for more than two rounds lead to slightly more improvements, at the cost of more computational time. The time of running both programs for two rounds are reported in Tables 1 and 2.

Sparc scales well to larger datasets; we show here the performance of Sparc and PBdagcon on a larger 20× PacBio *A. thaliana* dataset (genome size 120 Mbp). The longest backbone generated by DBG2OLC is 7.1 Mbp. Sparc finishes with 1/5th time and memory compared with PBdagcon while producing more accurate results. Here a pure PacBio full genome assembly generated by MHAP (*Berlin et al., 2015*) is used as the reference to calculate the error rates.

On an Oxford Nanopore dataset, we set $k = 2$, $g = 2$ and run the consensus algorithms for four rounds in consideration of the higher error rate. The per base error rates after 1/2/4 rounds are reported as Err1/2/4 in Table 3. The results using only Oxford Nanopore (ON) data and using hybrid data are reported in rows 1, 2 and rows 3, 4 respectively. With Oxford Nanopore data, Sparc obtains significantly lower error rate using hybrid data.

**Table 3  Results on an *E. coli* dataset using Oxford Nanopore sequencing.**

| Program | Coverage | N50 | # | Time | Memory | Err1 | Err2 | Err4 |
|---|---|---|---|---|---|---|---|---|
| Sparc | 30× ON | 4.61 MB | 1 | 2.3 m | 1.89 GB | 11.96% | 9.22% | 7.47% |
| PBdagcon | 30× ON | 4.61 MB | 1 | 10.0 m | 8.38 GB | 13.70% | 12.96% | 12.86% |
| Sparc | 30× Hybrid | 4.61 MB | 1 | 3.3 m | 1.86 GB | 0.72% | 0.59% | 0.46% |
| PBdagcon | 30× Hybrid | 4.61 MB | 1 | 13.2 m | 9.56 GB | 11.20% | 10.01% | 9.96% |

**Table 4  Memory and quality comparisons using different $k$, $g$ values.**

| K | g | Time | Memory | Error rate |
|---|---|---|---|---|
| 1 | 1 | 43 s | 2.3 GB | 0.16% |
| 2 | 1 | 55 s | 3.5 GB | 0.14% |
| 1 | 2 | 59 s | 1.6 GB | 0.18% |
| 2 | 2 | 68 s | 2.3 GB | 0.13% |

**Table 5  Performance of using different $b$ values.**

| Dataset | b | Err1 | Err4 |
|---|---|---|---|
| 30× PB Hybrid *E. coli* | 0 | 0.34% | 0.05% |
| 30× PB Hybrid *E. coli* | 5 | 0.17% | 0.02% |
| 30× PB Hybrid *E. coli* | 10 | 0.11% | 0.02% |
| 30× PB Hybrid *E. coli* | 15 | 0.08% | 0.02% |
| 30× ON Hybrid *E. coli* | 0 | 6.87% | 6.69% |
| 30× ON Hybrid *E. coli* | 5 | 0.88% | 0.70% |
| 30× ON Hybrid *E. coli* | 10 | 0.72% | 0.46% |
| 30× ON Hybrid *E. coli* | 15 | 0.69% | 0.48% |

Below 0.5% error rate is reached even though the raw error rate could be as high as 40%. In contrast, non-hybrid consensus generated less usable results due to this excessive error rate. The longest backbone in this test is 4.6 Mbp. The time of running both programs for four rounds are reported in Table 3.

Sparc is relatively insensitive to different parameters and hence easy to use even for inexperienced users. We vary the parameters in the second round of the 30× PacBio *E. coli* dataset using PacBio data only. The memory, time and quality of using different $k, g$ are reported in Table 4. Using a slightly larger $k$-mer size increases the per-base accuracy, as more strict matches are enforced, this effect is more significant on multi-ploid genomes, as different strands will have more distinct representations. However, this also increases the memory consumption because more branching nodes are created. Setting a larger $g$ helps to reduce the memory consumption. The influence of various amount of weight increase ($b = 0, 5, 10, 15$) to NGS data in the hybrid consensus setting can be found in Table 5. Empirically this parameter can be safely set as $b = 5 \sim 15$ without compromising the accuracy. In practice we would tentatively set it to be a low values ($b = 5 \sim 10$) to more sufficiently utilize the 3GS data.

## CONCLUSION

Consensus module is a critical component in the Overlap-Layout-Consensus assembly framework. With the introduction of 3GS technology, its importance is further raised. In this work we demonstrated a simple but efficient consensus algorithm that uses $k$-mers as building blocks and produces high-quality consensus from a position-specific $k$-mer graph. It supports hybrid sequencing data which leads to the significant improvements in both cost efficiency and computational efficiency. The proposed method is expected to significantly expand its applications in error correction and variant discovery. The consensus quality can also be enhanced further by incorporating platform specific, signal-level information.

## ACKNOWLEDGEMENTS

We thank Chris Hill and Jue Ruan for help and fruitful discussions.

### Funding

The research received funding from the following sources: NSFC (Grant No: 61175071 & 71473243) and ''Exceptional Scientists Program of Yunnan Province, China.'' The funders had no role in study design, data collection and analysis, decision to publish, or preparation of the manuscript.

### Grant Disclosures

The following grant information was disclosed by the authors:
NSFC: 61175071, 71473243.
Exceptional Scientists Program of Yunnan Province, China.

### Competing Interests

The authors declare there are no competing interests.

### Author Contributions

- Chengxi Ye conceived and designed the experiments, performed the experiments, analyzed the data, wrote the paper, prepared figures and/or tables, reviewed drafts of the paper.
- Zhanshan (Sam) Ma conceived and designed the experiments, wrote the paper, reviewed drafts of the paper.

### Data Availability

    The research in this article did not generate any raw data.

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
