# Peer review of "Sparc: a sparsity-based consensus algorithm for long erroneous sequencing reads"

_PeerJ, doi:10.7717/peerj.2016_

## Round 0.1 · original submission · Major Revisions

Both reviewers have questions and concerns about many technical details of the proposed method. Please clearly specify these details in the revised manuscript, and improve the writing.

·

Basic reporting

No Comment

Experimental design

What was the Illumina coverage placed into SparseAssembler for your hybrid consensus calling? How were these contigs grouped with their corresponding DBG2OLC contigs backbones from which reads were selected?
It's unclear if you only selected the longest contig(s) per experiment for dnadiff comparison. If you used the full collection of backbones, it'd be interesting to see a few extra metrics like N50 and number of contigs
Since you used the DBG2OLC output as your backbone and sequence selection, were there any regions of the reference that were not covered, and therefore polishing was not performed? I'd like to see assurance that we're not seeing a comparison of the two programs over only the 'simpler' (i.e. no repeats like STRs or homopolymer runs) - though given the positional information used, I wouldn't expect much change.

Validity of the findings

No Comments

Additional comments

In Table 1, what is the unit for Time?

Reviewer 2 ·

Basic reporting

1. English grammar could be improved
2. The testing dataset should put be placed in a more stable public location than in an individuals dropbox account
3. From the introduction it is unclear from the discussion whether the authors are discussing de novo assembly or reference-based assembly (where reads are mapped to a reference genome), because of the reference to a "target genomic region".

Experimental design

1. The Sparc methodology is not described in sufficient detail, pseudocode or additional text is needed
1a. The details on how "we reduce the edge weight by a small amount" for the "sparse penalty" are not provided
1b. It is unclear how "If a query region perfectly aligns to an existing region in the graph, we merge all the nodes and edges in the region". Merging of nodes and edges is not shown in Figure 1. It is unclear whether just a single query that aligns perfectly allows for merging of nodes, or if nodes are merged only after considering all queries.
1c. Is the breadth-first search algorithm guaranteed to find the heaviest path? Don't you have to backtrack all the way to the start node?
1d. Is it possible to have cycles in this graph or is it always acyclic?
1e. It is unclear how k and g affect the accuracy of the method since the alignments are provided as input. Please explain.

Validity of the findings

1. Sparc has a number of parameters (e.g., k, g, and the edge weight increase value for Illumina contigs) and the authors set these parameters (often in different ways) for their experiments without any justification. Therefore, the reader is left wondering whether the authors selected the parameter values that optimized its accuracy on the various test sets. In practice, one does not know the truth, and therefore parameters have to be set a priori. In particular, the performance benefits of Sparc appear most prominent in the "Hybrid" case, for which the larger edge weight parameter for Illumina contigs is very important and the choice of "5" for this is unjustified.

---

## Round 0.2 · Major Revisions

From the three reviewers' comments on the revised manuscript, there are still serious concerns about the clarity of terms, the methodology and the results. Important comments are regarding the construction of the k-mer graph and the justification of the hybrid approach. Please carefully address the reviewers' comments point by point.

Reviewer 2 ·

Basic reporting

No comments

Experimental design

1. Section 2.2: The word "merging" is a bit confusing - this suggests that you have created a k-mer graph for each read and are merging the common nodes with the backbone graph, but you don't say anything about creating k-mer graph for each read. Please clarify the meaning of "merging" here.

2. Section 2.3: Please clarify what "reduce" means in adjusting the weights of the graph. Is this subtraction or some sort of downscaling? What is the "coverage"? Is it the initial weight of the edge?

Validity of the findings

No comments

Additional comments

The contribution of Sparc could be made more clear if the performance gains of Sparc over PBdagcon can be explained by differences in the underlying methodologies. In particular, it appears that Sparc's main advantage is in the hybrid case. Is this simply because of the increased edge weight parameter for NGS reads (the "b" parameter)? If "b" is set to 1, does Sparc become similar to PBdagcon in terms of performance? It also appears possible from the results presented that higher values of the "b" parameter results in faster convergence (in terms of the number of rounds required until the assembly is stable). Is it simply that Sparc converges faster than PBdagcon and that if both are run to convergence they will give the same accuracy? Since the programs do not take much time to run, I recommend that they be run for more than two rounds to see what level of error they converge to.

Reviewer 3 ·

Basic reporting

The authors seem to rely sometimes on "common knowledge", however, in these cases the readability of the manuscript could be improved by providing references or explanation of the specific term. In particular it would be helpful to:
- spell out what "hybrid" data set means,
- provide reference for the current PacBio and Oxford Nanopore error rates.

There are also some spelling errors and typos which should be corrected.

Experimental design

no comments

Validity of the findings

no commenst

Additional comments

Taking into account the reviewers suggestions has improved the quality and readability of the manuscript. There is just one more issue:
Authors claims in the introduction that: "Experiments with Sparc show that our algorithm can efficiently provide high-quality consensus sequences with error rate <0.5% using both PacBio and Oxford Nanopore sequencing technologies." what is of course true based on tables 1-3. The impression is however, that a such improvement can be done for PacBio or Nanopore data only, while almost in all cases a such low error rate has been obtained for hybrid approach. This should be clear up.

Reviewer 4 ·

Basic reporting

The paper presents consensus method for long reads. The method is designed for '3rd generation' sequencing platform from Pac Bio and Oxford Nanopore Technologies.

Paper is missing a clear statement of the problem and motivation of the proposed method. Additionally it contains extensive number of sentences which is hard to follow.
Thus the paper needs modification to improve readability. Paper might benefit from a schematics figures explaining the problem, and the differences from between the consensus and assembly problem, as well as the influence of the short and long reads.

For example the formulation of assembly problem is misleading (line 29). The goal of the assembly is to produce the original genome (set of chromosomes),. Due to short read length and repetitive structure of genome, assembly tool often are not able to produce a single genome linage instead producing contiguous sequences(contigs). Additionally clear explanation of the difference between the assembly and consensus problems might help reader. A schematics figure might be considered as a possibility. The review paper 'Sequence assembly demystified' from Nature Genetics migth be useful.



Minor :

1) line 7. The sentence stating that assembly is a 'Fundamental Computation task' is confusing and needs more details. It is not clear why assembly is 'Fundamental Computation task' and not a fundamental problem in biology or genomics. A

2) The explanation of the algorithm in 2.2 is confusing. It would be better if the authors clarify the position specific k-mer graphs.

3) The manuscript is written too informally, such as too many "we did something" in the method.

4) There are some minor grammatical errors

Experimental design

The proposed methods was examined on various datasets of the contemporary long reads.

Validity of the findings

As the problem is not clear stated is it impossible to judge the 'Validity of the Findings'

---

## Round 0.3 · Minor Revisions

Please address the comments from Reviewers 2 and 3 below. Please improve the readability and clarity of the manuscript.

Reviewer 2 ·

Basic reporting

No Comments.

Experimental design

No Comments.

Validity of the findings

In their response to my previous comments about explaining the performance gains of Sparc over PGdagcon in terms of the methodological differences, the authors stated that if the critical parameter b is set to one (and k=g=1), then Sparc and PBdagcon are similar in accuracy. However, no new results are provided in the paper regarding this point, which I believe is an important one. I strongly recommend that the authors provide the results obtained with Sparc with b=1 in Tables 1, 2, 3. This explains a large amount of the performance gain of Sparc.

In addition, in the paper and response is states that, for the PacBio datasets, additional rounds do not improve the results. It is unclear if this (a) was for both Sparc and PBdagcon and (b) was also for the PacBio Hybrid data sets. To make things clear and to let the data speak for themselves, I recommend adding an addition column to the various tables (Err3), which gives the results after three rounds.

Reviewer 3 ·

Basic reporting

Although, the authors have improved the readability of the text, I still have difficulties to fully follow the logic especially in the introduction. It is better in the rest of the Manuscript, however, also there I found at least one problematic point. At the beginning of the method section authors wrote: ‘Sparc consists of the following four simple steps (Fig. 2): …’, where the description of these four steps perfectly match the pseudo-code from the insert. But Fig. 2 seems to be ‘A toy example of constructing the position specific sparse k-mer graph.’ from the next page. What in fact is/suppose to be a Fig. 2 here?

In general, the manuscript should be carefully checked and its readability should be further improved.

Experimental design

"No Comments"

Validity of the findings

"No Comments"

Reviewer 4 ·

Basic reporting

Authors addressed all the comments. Paper can be accepted

Experimental design

Authors addressed all the comments. Paper can be accepted

Validity of the findings

Authors addressed all the comments. Paper can be accepted

---

## Round 0.4 · accepted · Accept

Before we publish your manuscript, please provide the specific license information about your open source software.

Reviewer 2 ·

Basic reporting

No Comments

Experimental design

No Comments

Validity of the findings

No Comments

Reviewer 3 ·

Basic reporting

No Comments

Experimental design

No Comments

Validity of the findings

No Comments

Additional comments

Authors have addressed all the previous comments.